

# Eyestalk transcriptome analysis reveals sexually dimorphic host-parasite interactions: divergent molecular strategies of *Polyascus* sp. for reproductive and growth regulation in *Eriocheir sinensis*

Jing Xie[1,2], Congping Ying[2], Zhen Tang[2], Yanping Yang[2] and Kai Liu[1,2]

[1] Wuxi Fisheries College, Nanjing Agricultural University, Wuxi, China
[2] Key Laboratory of Freshwater Fisheries and Germplasm Resources Utilization, Ministry of Agriculture and Rural Affairs, Freshwater Fisheries Research Center, Chinese Academy of Fishery Sciences, Wuxi, Jiangsu, China

## ABSTRACT

**Background**. The Chinese mitten crab (*Eriocheir sinensis*), an economically critical migratory species inhabiting both freshwater and marine environments, is susceptible to parasitization by the rhizocephalan barnacle *Polyascus* sp. in estuarine ecosystems. In this study, *E. sinensis* was found to be parasitized by *Polyascus* sp. in the freshwater waters of the Anqing section of the Yangtze River, which manifested itself as restricted growth and development, hindered gonadal development, and apparent changes in sexual characteristics.

**Methods**. Morphological analysis, serological testing, and eyestalk transcriptome analysis were conducted on male and female parasitized and non-parasitized crabs to investigate the molecular mechanisms by which *Polyascus* sp. affects host growth and reproduction.

**Results**. The study revealed that parasitized hosts of both sexes were significantly smaller than normal individuals and exhibited gonadal atrophy. Male hosts displayed feminization of secondary sexual characteristics, whereas females showed no significant alterations. Testosterone and ecdysterone levels decreased in male hosts, whereas estradiol levels declined in female hosts. Transcriptomic profiling revealed differentially expressed genes predominantly associated with metabolic processes, stimulus response, signal transduction, and reproduction. Thirteen key candidate genes related to parasite-induced suppression of host molting and growth, fourteen key candidate genes involved in male reproductive regulation, and ten candidate genes affecting female gonadal development were identified. Key genes involved in gonadal development—including *5-HT1B, CHH, MIH, JHE1*, Dmrt family, cytochrome P450 family, *Abd-A*, and *Antp* exhibited sex-specific expression patterns. Additionally, critical growth-related genes such as *Dib*, *HR38*, and *Cht3* exhibited significant alterations in hosts of both sexes. *Polyascus* sp. parasitism suppresses gonadal development and growth in both sexes, employing distinct regulatory strategies. It uses a "neuroendocrine disruption and reproductive gene regulation" strategy to regulate reproductive development in male hosts, while influencing female reproductive metabolism through a

Corresponding author
Kai Liu, liuk@ffrc.cn

"Nutrient Hijacking-Hormonal Suppression" mechanism. Additionally, the parasitism disrupts host growth *via* "molting inhibition". These findings elucidate the sex-specific metabolic, reproductive, and growth disruption strategies of *Polyascus* sp., providing new insights into host-parasite interactions in decapod crustaceans.

## INTRODUCTION

The Chinese mitten crab, *Eriocheir sinensis*, is a crustacean widely distributed in coastal and connected estuarine ecosystems. As a catadromous species, it spends the majority of its life in freshwater habitats before migrating to marine environments for reproduction upon reaching sexual maturity (*Chen & Du, 2017*). The life cycle of *E. sinensis* progresses through several developmental stages: fertilized eggs, zoea, megalopa, juvenile crabs, young crabs, and mature crabs (*Cui et al., 2021*). In the Yangtze River basin, this species exhibits a catadromous life history pattern similar to cultured populations. Reproductive activities occur in estuarine brackish waters (salinity 15–23) from January to April (*Geng, Zhang & Zhao, 2018*). After spawning, embryogenesis typically takes 3–4 months before hatching into stage I zoea. These zoea undergo five molting cycles (lasting 30–40 days) to reach the megalopa stage, subsequently migrating toward lower salinity zones (*Wang et al., 2016*). The megalopa molts once more to develop into stage I juvenile crab, which then undergoes approximately ten successive molts to gradually mature into a young crab. In freshwater habitats, the crab completes five growth molts followed by one reproductive molt before returning to brackish estuarine waters for mating and spawning, thereby completing its life cycle. Throughout its ontogeny in the Yangtze River system, *E. sinensis* typically completes 21 molting cycles (*Jie, 2005*). Whether through excessive or insufficient molting events during juvenile or adult developmental stages, it results in substantial impairments to both growth and fattening (*Wang et al., 2016*).

In China, *E. sinensis* is a commercially important aquaculture species due to its wide distribution and high fattening potential. However, its cultivation is significantly impacted by various diseases, particularly parasitic infections such as Sacculinosis, which cause substantial economic losses in aquaculture. *Polyascus* sp. is a genus of parasitic crustaceans within the Cirripedia, Rhizoeephala, Sacculinidae, which specialize in parasitization of crabs and some other crustaceans (*Glenner, 2001*; *Walker, 2001*; *Glenner & Hebsgaard, 2006*; *Waiho et al., 2017*). Like other rhizocephalan barnacles, *Polyascus* sp. exhibits a complex life cycle that occurs in seawater, involving several distinct developmental stages (*Korn, Rybakov & Kashenko, 2000*; *Walker, 2001*). The released nauplius larvae undergoes a series of molts into cypris larvae, wherein the female cypris larvae attaches to the host's cuticle and develops an extensive root-like system (interna) that permeates along the host's intestines until other tissues for nutrient absorption, eventually forming an external reproductive structure (externa). After that, the male cypris larvae implants into the

externa for gamete fusion. Upon maturation, the externa releases new nauplius larvae into the aquatic environment, thus completing the parasitic life cycle.

Rhizocephalan barnacles are known to disrupt host physiology by inhibiting molting, reproduction, and immune function, effectively diverting host resources to support their own growth and reproduction (*Li et al., 2011*; *Belgrad & Griffen, 2015*; *Hsiao et al., 2016*; *Waiho et al., 2017*). These parasites alter host metabolism through nutrient depletion, transferring absorbed nutrients *via* rootlets to the externa for their development (*Hsiao et al., 2016*). Additionally, sacculinid rhizocephalans significantly impact host endocrine systems, including reduced ecdysteroid levels (*Andrieux et al., 1976*) and diminished neurosecretory substance storage in the sinus gland (*Rubiliani & Payen, 1979*). Earlier studies demonstrated that haemolymph from parasitized crabs induces feminization of male hosts, testicular regression, molting suppression, and behavioral changes that halt reproductive activity (*Andrieux, Herberts & Frescheville, 1981*).

Recent advances in high-throughput sequencing and multi-omics approaches (*e.g.*, transcriptomics, proteomics, metabolomics) have transformed parasitology research, enabling in-depth analysis of infection dynamics and host-parasite interactions (*Cantacessi et al., 2012*; *Li et al., 2013*; *Patino & Ramírez, 2017*). For instance, *Zhao et al. (2023)* used hepatopancreas transcriptomics and proteomics to elucidate *Polyascus* sp. effects on *E. sinensis*, while Waiho et al. employed gonadal transcriptomics to identify the differential expression of genes associated with reproduction, immune response, and growth in the gonads of Scylla olivacea following rhizocephalan infection (*Waiho et al., 2020*). As an important neuroendocrine organ in crustaceans, the eyestalk regulates molting, reproduction, and metabolism through neurohormone synthesis (*De Kleijn & Van Herp, 1995*; *Li et al., 2025*). Previous studies have leveraged eyestalk transcriptomics to investigate precocity in *E. sinensis* (*Xu et al., 2015*) and post-molt physiological regulation (*Toyota et al., 2023*).

While rhizocephalan parasitism systemically affects host tissues, including the hepatopancreas, hemolymph, gonads, muscles, and ganglia (*Rowley et al., 2020*), most omics studies have focused on the host's hepatopancreas, gonads, and hemolymph. Meanwhile, the eyestalk serves as a crucial neuroendocrine target for examining how parasites alter host endocrine function and physiological processes in crustaceans. In this study, we analyzed the transcriptome of the eyestalks of *E. sinensis* following parasitization by *Polyascus* sp. using RNA sequencing technology, aiming to investigate the effects of *Polyascus* sp. parasitism on the gene expression profiles of *E. sinensis* eyestalks, identify key genes and pathways related to neuroendocrine regulation, and provide a scientific foundation for understanding how *Polyascus* sp. influences the reproductive development and growth regulation of *E. sinensis*. The findings from this research will offer new insights into the molecular interactions between Rhizoeephala parasites and their hosts, as well as provide a theoretical framework for the prevention and treatment of *Polyascus* sp. in *E. sinensis*.

## MATERIALS AND METHODS

### Sample collection

During October to December 2023, a flood season survey of Chinese mitten crabs was conducted in the Anqing section of the Yangtze River mainstem. The survey operation waters were from the lower net at Anqing Petrochemical Wharf (117°01′22.89″E, 30°29′51.07″N) to the upper net at Anqing Yangtze River Bridge (117°05′10.76″E, 30°30′08.63″N) in the main stream of the Yangtze River. Fishing in this study was carried out under the relevant laws and regulations, with the field permit No. 00002582087 (Data S1). All parasitized and non-parasitized crabs were assayed for representative biological indices (*Jing et al., 2024*). Complete eyestalks of parasitized and non-parasitized crabs were cut and collected, quick-frozen in liquid nitrogen, and then refrigerated at −80 °C. Unilateral eyestalks of three parasitized male crabs were taken as the male parasitized group (T1), unilateral eyestalks of three parasitized females were taken as the female parasitized group (T2), unilateral eyestalks of three normal males were taken as the male normal control group (CK1), and unilateral eyestalks of three normal females were taken as the female normal control group (CK2) for transcriptome assay analysis. At the same time, serum samples were collected from the third appendage of crabs and stored at −80 °C for subsequent testing.

### RNA acquisition, library construction and sequencing

Eyestalk samples were stabilized at −80 °C within 1 month and used for RNA extraction. Total RNA from 12 samples was extracted using Trizol reagent. All samples had high integrity, as indicated by results from the Nanodrop 2000 (ThermoFisher, CA, USA) and Agilent 4200 (Agilent Technologies, CA, USA). RNA extraction, library construction, and sequencing were performed by Gene Denovo Biotechnology Co., Ltd. (Guangzhou, China). The statistical power of T1 *vs.* CK1 was calculated with RNApower (R command) as 0.99 (depth = 50, cv = 0.0042, effect = 2, biological replicates = 3, alpha = 0.05), and 0.99 (depth = 50, cv = 0.0462, effect = 2, biological replicates = 3, alpha = 0.05) of T2 *vs.* CK2. Referring to Hieff NGS® Ultima Dual-mode mRNA Library Prep Kit (12309ES, Yeasen, Shanghai, China), the extracted mRNA was enriched using mRNA capture beads. The purified and fragmented mRNA was used for subsequent cDNA synthesis, followed by end repair, A-tailing, Adapter ligation, and polymerase chain reaction (PCR) amplification. The final amplification product was sequenced on Illumina Novaseq X Plus. The raw data were subjected to quality control procedures to eliminate low-quality reads. Specifically, reads containing adapter sequences, those with an N base ratio exceeding 10%, reads composed entirely of adenine bases, and reads in which more than 50% of the bases had a quality score of $Q \leq 20$ were removed. This filtering process resulted in a dataset of high-quality clean reads (*Chen et al., 2018*). All downstream analyses were conducted on the cleaned data.

### Differential expression and enrichment analysis

The resulting clean reads were mapped to the reference genome using HISAT2 2.1.0. Gene expression levels were quantified in Fragments Per Kilobase of transcript per Million

**Table 1** Primers of detected genes for qPCR.

| Gene | Forward primer (5′-3′) | Reverse primer (5′-3′) | Length (bp) |
|---|---|---|---|
| PGDS | GAGTGGTTCATCGGAGAC | TGGGAGGTTCTGGACTTT | 141 |
| CYP2L1 | GCAGATGGTGGCAAGTAA | CTCGTAGAAGTAGGTGAAGAA | 217 |
| GST | CATTGTTCCGTGGAGAGG | TGTTGGTGTGCTTGCTTA | 192 |
| HR38 | CCTATGTCCTCCGCAATC | TGTTCTGACCACTTCCTTC | 279 |
| Dib | CTGAAGGCGGTGATGAAG | AGGCAGGAAGTGTAGAGG | 183 |
| β-actin | GCGAGACATCAAGGAAAAGC | CGTCAGGGAGCTCGTAAGAC | 106 |

mapped reads (FPKM), and gene read counts were obtained through HTSeq-count (*Pertea et al., 2016*; *Chen et al., 2018*). Principal component analysis (PCA) was conducted using R (http://www.r-project.org/) to assess the biological replicability of samples. Differential expression analysis was performed with DESeq2 software (*Love, Huber & Anders, 2014*), applying a significance threshold of Q value < 0.05 and $|\log_2 FC| \geq 2$ for differentially expressed genes (DEGs). Gene Ontology (GO) enrichment analyses (*Ashburner et al., 2000*) of DEGs were conducted using GO.db (3.14.0) to identify significantly enriched terms, and Kyoto Encyclopedia of Genes and Genomes (KEGG) database was used to further elucidate the biological functions of the identified genes (*Kanehisa & Goto, 2000*).

## Detection of estrogen, testosterone, and ecdysterone in serum with ELISA

According to the method provided by the manufacturer (SBJ-CR0120-48T, SBJ-CR0121-48T, SBJ-CR0122-48T, SenBeiJia Biological Technology Co., Ltd., Nanjing, China), the absorbance (OD value) of the wells in which the test samples were detected was measured spectrophotometrically at a wavelength of 450 nm, and the contents of crab estradiol (E2), testosterone (T) and ecdysterone in the samples were calculated by standard curve calculation (Data S2).

## qPCR validation

Total RNA was extracted from the eyestalks according to the method described above and then reverse transcribed into first-strand complementary DNA (cDNA) using a HiScript III RT SuperMix for qPCR (+gDNA wiper) (R323-01, Vazyme, Nanjing, China). Five DEGs were selected: prostaglandin D synthase (*PGDS*), cytochrome P450 CYP2 (*CYP2L1*), delta glutathione S-transferase GST (*GST*), probable nuclear hormone receptor HR38 isoform X2 (*HR38*), and CYP302a1 (*Dib*). β-actin served as the reference gene. The primers used to amplify the genes were designed by Primer Premier 6.0 and are shown in Table 1. The qPCR assay was performed on a Bio-Rad CFX96 touch Real-Time PCR Detection System (Bio-Rad, Hercules, USA). The real-time quantitative PCR was performed with AceQ qPCR SYBR Green Master Mix kit (Q121-01, Vazyme, Nanjing, China). All reactions contained three replicates to ensure the reliability and reproducibility of results. PCR conditions were as follows: 95 °C for 300 s, followed by 40 cycles at 95 °C for 10 s, 60 °C for 30 s, and one cycle at 95 °C for 15 s, 60 °C for 60 s, 95 °C for 15 s. The fold changes of genes detected by qPCR were analyzed with $2^{-\Delta\Delta Cq}$.

## Data processing and statistics

Statistical analysis and data visualization were performed using GraphPad Prism 10.0 (GraphPad Software, San Diego, CA, USA). An independent sample "$t$" test with Welch's correction method was applied to compare the data of the groups. Statistical significance was defined as $P < 0.05$.

## RESULTS

### Biological changes in *E. sinensis* after parasitization by *Polyascus* sp.

Compared to normal crabs, parasitized crabs exhibited significantly reduced average weight and carapace width, while all parasitized crabs had a higher accumulation of sediment and several white capsules in the abdomen (published) (*Jing et al., 2024*). Based on the prior findings of the research team, each parasitized crab exhibits a varying number of externas on its abdomen (the number of externas ranged from 1 to 27), and certain individuals display black scars on their abdomens, which remain following the detachment of the externas. In male crabs of the same size range (carapace width between 50 and 60 mm), parasitic infection induced the growth of setae on the abdomen and the development of female-like pleopods, along with a reduction in chela hair, collectively indicating feminization (Figs. 1A and 1C). Additionally, the testes and accessory glands showed marked atrophy and degeneration. Compared to non-parasitized female crabs of equivalent size, the parasitized females exhibited ovarian atrophy and reduced chela size in some individuals. Variations in chela setation (both increased and decreased density) were observed, while neither the number nor morphology of pleopods showed significant alterations (Figs. 1B and 1D). Enzyme-linked immunoassay (ELISA) analysis of serum samples revealed significant endocrine alterations in parasitized crabs. In males, the parasitized group exhibited markedly decreased testosterone levels compared to normal males ($P = 0.046$, $P < 0.05$), while estradiol levels remained unchanged and ecdysterone levels were reduced ($P = 0.013$, $P < 0.05$). Conversely, parasitized females showed significantly lower estradiol levels relative to normal females ($P = 0.018$, $P < 0.05$), with no significant changes observed in testosterone or ecdysterone levels (Figs. 1E–1G).

### Identification of DEGs

Transcriptomic analysis was conducted on eyestalk samples, with sample correlations initially assessed through principal component analysis (PCA). The PCA revealed clear segregation among four experimental groups: parasitized males, parasitized females, normal males, and normal females (Fig. 2A). Differential gene expression was determined using stringent criteria ($|\log2FC| \geq 2$ and FDR $< 0.05$). Comparative analysis identified 4,180 differentially expressed genes (DEGs) in T1 *vs.* CK1, comprising 2,721 up-regulated and 1,459 down-regulated genes (Data S3). Similarly, 1,338 DEGs were detected in T2 *vs.* CK2, with 819 up-regulated and 519 down-regulated genes (Fig. 2B) (Data S4). A Venn diagram analysis of the two comparison groups (Fig. 2C) revealed 617 DEGs specifically associated with *Polyascus* sp. parasitization (Data S5).

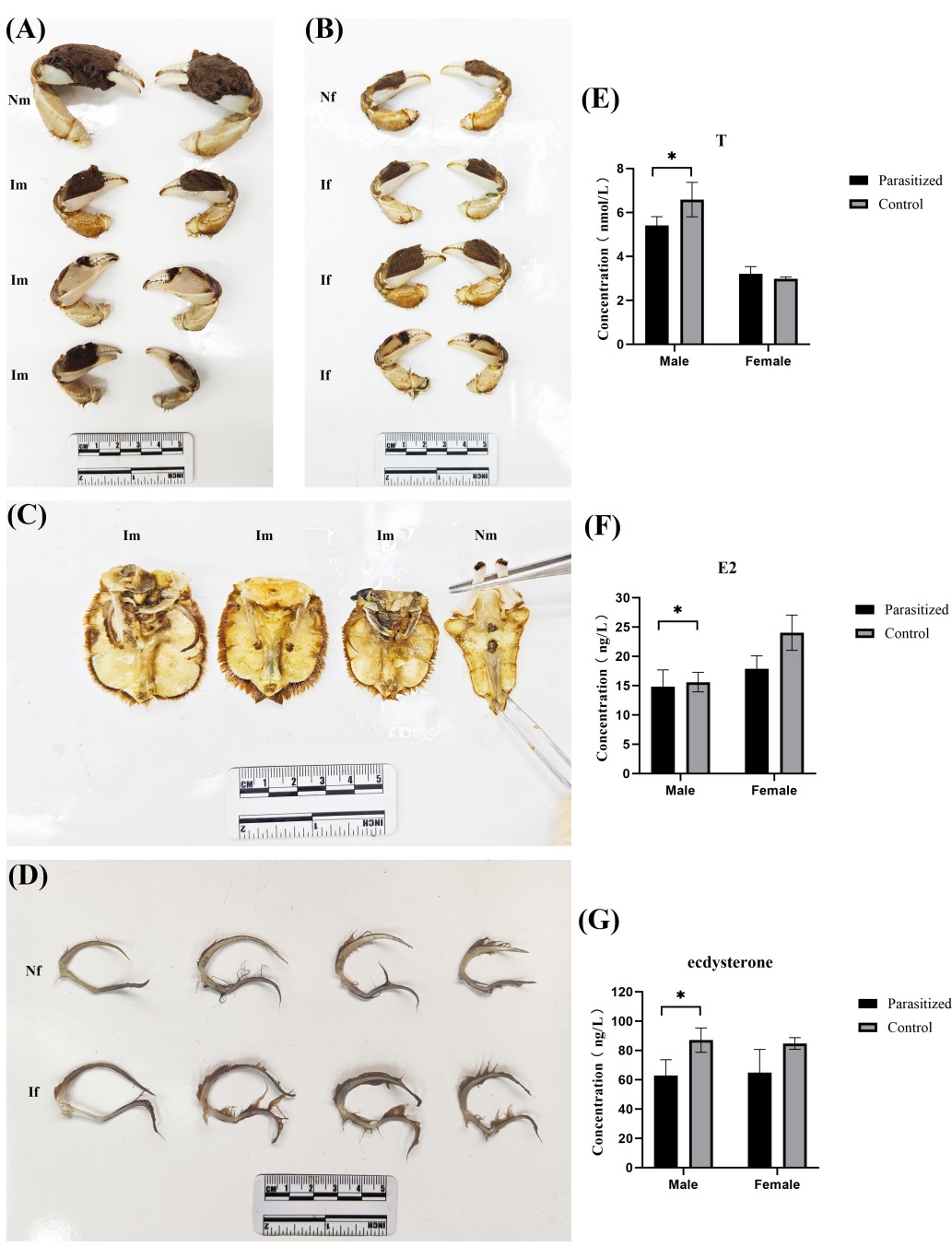

**Figure 1** Biological changes in the *E. sinensis* after parasitization by *Polyascus* sp. (A) Comparison of cheliceraes in parasitized and normal males. (B) Comparison of cheliceraes in parasitized and normal females. (C) Comparison of abdominal limbs of parasitized and normal males. (D) Comparison of the abdominal limbs of parasitized and normal females. (E–G) Testosterone, estrogen, and ecdysterone in serum tested by ELISA. Photo credit: Jing Xie.

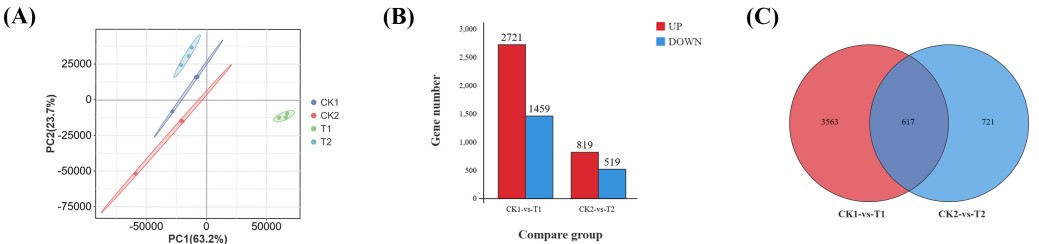

**Figure 2  Identification of differential genes (DEGs).** (A) Principal component analysis (PCA) of four groups of eyestalk samples. (B) Histogram of DEGs. (C) Venn diagram of DEGs for two comparative transcriptomes.

## Functional enrichment analysis

To elucidate the biological functions of DEGs in eyestalks between parasitized and non-parasitized crabs of different sexes, we performed comprehensive GO term and KEGG pathway enrichment analyses. Transcripts of DEGs were mapped to Gene Ontology (GO) terms across three categories: molecular function (MF), cellular component (CC), and biological process (BP). In the male parasitized *vs.* normal male comparison, DEGs were enriched in 53 GO terms, and the female parasitized *vs.* normal female comparison showed enrichment in 46 GO terms. The most significantly annotated terms among two comparison groups are both cellular process (GO:0009987), metabolic process (GO:0008152), binding (GO:0005488), catalytic activity (GO:0003824), and cellular anatomical entity (GO:0110165). Notably, both comparison groups exhibited significant enrichment in multiple BP terms related to response to stimuli (GO:0050896), signaling (GO:0023052), reproduction (GO:0000003), and reproductive processes (GO:0022414), demonstrating substantial differences in neuroendocrine regulation-related molecules between parasitized and normal crabs (Fig. 3A).

KEGG pathway analysis revealed that the 4,180 DEGs from T1 *vs.* CK1 were enriched in 346 pathways, while the 1,338 DEGs from T2 *vs.* CK2 were enriched in 297 pathways. These pathways were primarily categorized into six major classes: metabolism, human diseases, organismal systems, genetic information processing, cellular processes, and environmental information processing (Fig. 3B). Both comparison groups showed the highest gene enrichment in pathways including global and overview maps from metabolism class, cancer overview from human diseases, endocrine system from organismal systems, and signal transduction in environmental information processing, suggesting similar overall patterns of pathway regulation by *Polyascus* sp. parasitization.

Further analysis of the 617 common DEGs shared between T1 *vs.* CK1 and T2 *vs.* CK2 (Fig. 3C) revealed the proteasome (ko03050) pathway as the most significantly enriched, followed by ribosome biogenesis in eukaryotes (ko03008) and spinocerebellar ataxia (ko05017). Additionally, neurodegenerative disease-related pathways were enriched among the common DEGs, including Alzheimer's disease (ko05010), Parkinson's disease (ko05012), Huntington's disease (ko05016), and Neuroactive ligand–receptor interaction (ko04080). Among these pathways, up-regulated genes contain various proteasomes,

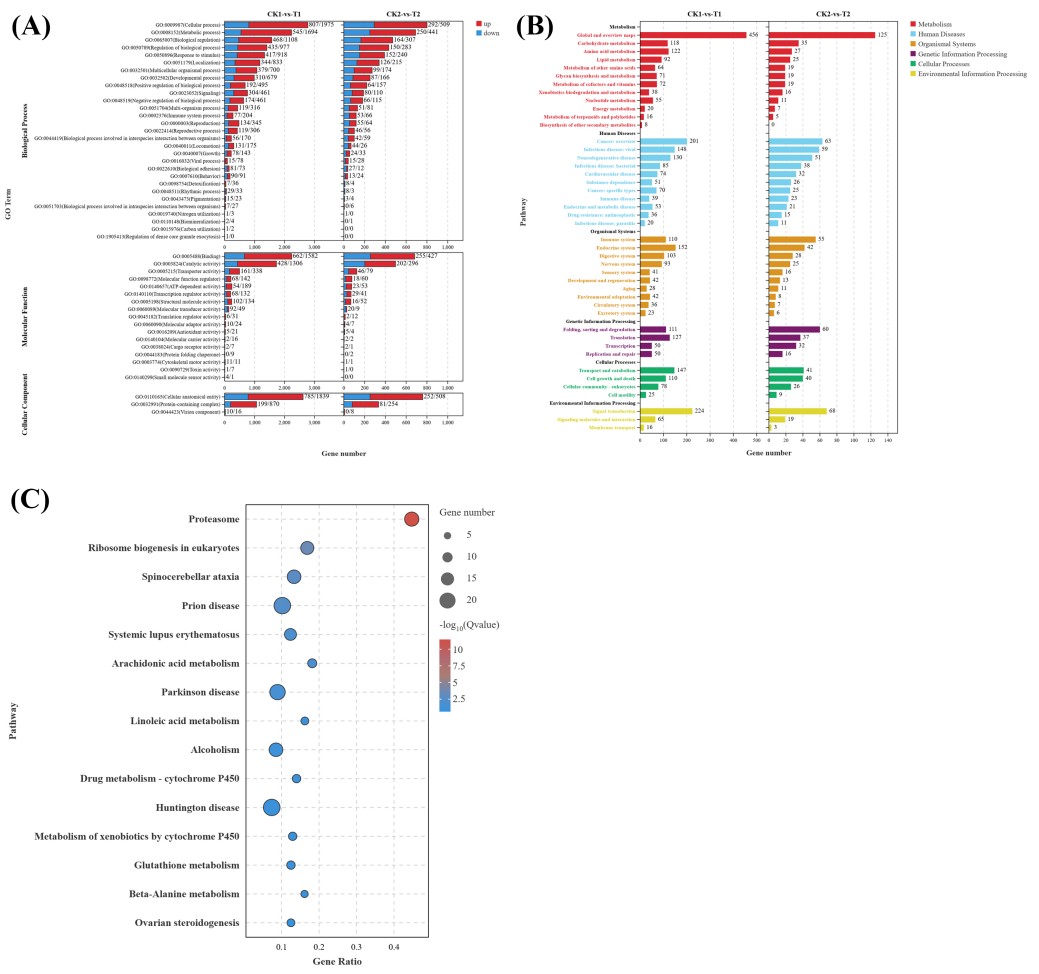

**Figure 3** **Gene ontology (GO) enrichment analyses and KEGG pathway enrichment analyses of DEGs in eyestalk transcriptome.** (A) GO enrichment analysis of all DEGs. (B) KEGG pathway enrichment analysis of all DEGs. (C) KEGG analysis of shared DEGs between T1 *vs.* CK1 and T2 *vs.* CK2.

and down-regulated genes contain cyclic adenosine monophosphate (AMP)-dependent transcription factor ATF-1-like isoform X1, Tubulin Alpha Chain-like, and other genes related to catalytic activity.

## Enrichment of growth-related DEGs

GO enrichment analysis of growth-related DEGs (Data S6) between the male parasitized group and the normal control group (Fig. 4A) revealed the enriched GO terms contained growth (GO:0040007), response to organic substance (GO:0010033), response to chemical (GO:0042221), anatomical structure development (GO:0048856), response to stimulus (GO:0050896), *etc.* In addition, KEGG analysis (Fig. 4B) showed that the top 20 significantly enriched pathways were primarily associated with the following KEGG B-class categories: Cell growth and death, including the cell cycle (ko04110), necroptosis (ko04217), ferroptosis (ko04216), and apoptosis (ko04210) pathways; cardiovascular diseases, such as hypertrophic cardiomyopathy (ko05410) and dilated cardiomyopathy (ko05414); signal

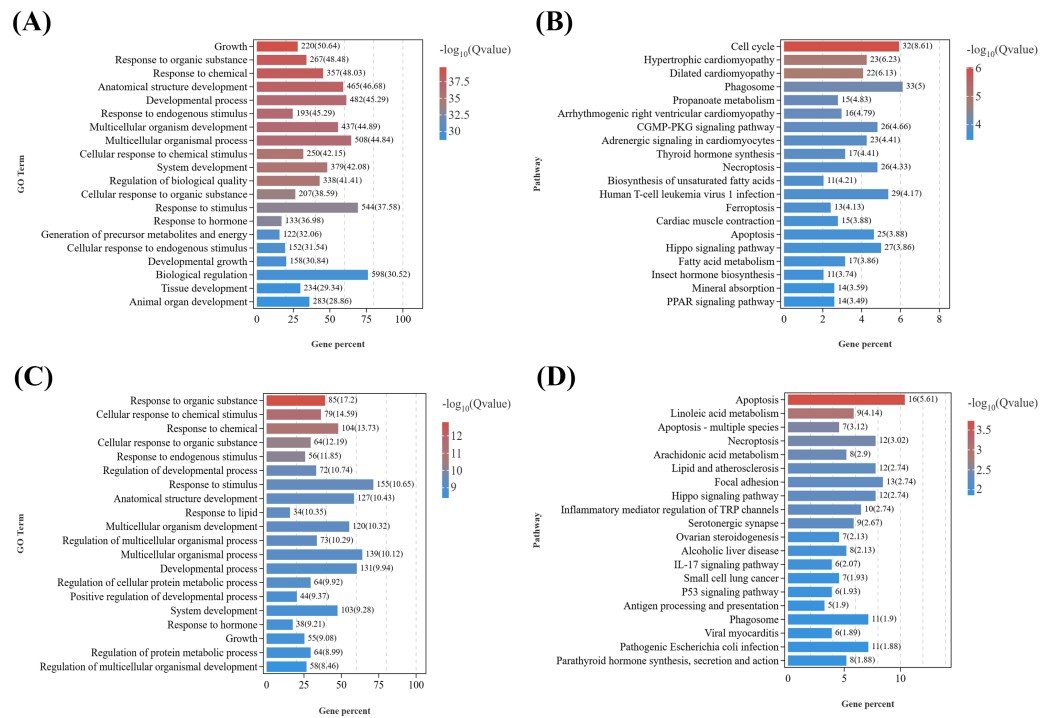

**Figure 4** **KEGG pathway enrichment analyses of DEGs related to reproduction.** (A) GO enrichment analysis of DEGs related to growth in males. (B) Top 20 KEGG pathways of DEGs related to growth in males. (C) GO enrichment analysis of DEGs related to growth in females. (D) Top 20 KEGG pathways of DEGs related to growth in females.

transduction, including the Hippo signaling pathway (ko04390) and cyclic guanosine monophosphate-dependent protein kinase (cGMP-PKG) signaling pathway (ko04022); Endocrine system, encompassing thyroid hormone synthesis (ko04918), peroxisome proliferator-activated receptors (PPAR) signaling pathway (ko03320), parathyroid hormone synthesis, secretion and action (ko04928), aldosterone synthesis and secretion (ko04925), and thyroid hormone signaling pathway (ko04919).

Similarly, based on the results of GO enrichment analysis, growth development-related DEGs (Data S7) from T2 *vs.* CK2 were enriched in GO terms including response to organic substance (GO:0010033), cellular response to organic substance (GO:0071310), response to endogenous stimulus (GO:0009719), multicellular organismal process (GO:0032501), *etc.* (Fig. 4C). KEGG enrichment analysis of (Fig. 4D) those DEGs demonstrated that the top 20 significantly enriched pathways included: cell growth and death, such as apoptosis (ko04210), necroptosis (ko04217), and cell cycle (ko04110); lipid metabolism, including linoleic acid metabolism (ko00591) and arachidonic acid metabolism (ko00590); signal transduction, such as the Hippo signaling pathway (ko04390) and tumor necrosis factor (TNF) signaling pathway (ko04668); Other signaling pathways, including focal adhesion (ko04510), serotonergic synapse (ko04726), parathyroid hormone synthesis, secretion and action (ko04928), and cytokine-cytokine receptor interaction (ko04060).

**Table 2** DEGs related to growth.

| Gene ID | Name | Change trend in DEGs of CK1 *VS* T1 | Change trend in DEGs of CK2 *VS* T2 | Description |
|---|---|---|---|---|
| ncbi_127000051 | *Nvd* | up | / | short-chain type dehydrogenase/reductase y4vI-like |
| ncbi_127004940 | *Dib* | down | down | cyp302a1 |
| ncbi_126999678 | *Chs-2* | down | / | chitin synthase |
| MSTRG.12858 | *Cht3* | up | up | chitinase-3-like protein 1 |
| ncbi_126987613 | *Cht5* | down | down | chitinase 5, partial |
| ncbi_126995807/ncbi_126996226 | *Cht10* | down | down | chitinase/probable chitinase 10 isoform X3 |
| ncbi_127004622 | *HR38* | down | down | probable nuclear hormone receptor HR38 isoform X2 |
| ncbi_127002945 | *HR4* | down | / | hormone receptor 4-like isoform X1 |
| ncbi_127003430 | *HR3* | down | down | probable nuclear hormone receptor HR3 isoform X1 |
| ncbi_127010442 | *Ftz-f1* | down | down | nuclear hormone receptor FTZ-F1 beta-like isoform X1 |
| ncbi_127002022 | *Ccap* | up | / | crustacean cardioactive peptide |
| ncbi_127009705 | *NO* | up | / | nitric oxide-associated protein 1-like isoform X1 |
| ncbi_126990297 | *Bursα* | up | | bursicon alpha subunit, partial |

Notably, several GO terms were commonly enriched in both male and female comparisons, including response to organic substance, response to chemical, response to endogenous stimulus, anatomical structure development, *etc.* It is indicated that parasitism of *Polyascus* sp. could significantly affect tissue development and signal transduction, which could specifically reflect on the formation of exokoleton in crustaceans (*Girish, Swetha & Reddy, 2017*; *Zhang et al., 2018*; *Knigge, LeBlanc & Ford, 2021*). Accordingly, thirteen key genes containing chitin synthase, chitinase, crustacean cardioactive peptide, *etc.*, were identified as candidate genes associated with the effects of *Polyascus* sp. parasitism on the host's growth (Table 2).

**Enrichment of reproduction-related DEGs**

Given that the DEGs in the T1 *vs.* CK1 were primarily enriched in GO terms related to metabolic processes, response to stimuli, signal transduction, and reproductive processes (Fig. 5A), considering the marked gonadal atrophy and abnormal secondary sexual characteristics (feminization) observed in parasitized males, we conducted KEGG enrichment analysis on reproduction-related DEGs (Data S8) from the T1 *vs.* CK1 (Fig. 5B). The results revealed 20 most significantly enriched pathways, including: ribosome biogenesis in eukaryotes (ko03008), fatty acid metabolism (ko01212), biosynthesis of unsaturated fatty acids (ko01040), insect hormone biosynthesis (ko00981), ovarian steroidogenesis (ko04913), oxytocin signaling pathway (ko04921), serotonergic synapse (ko04726), et al. Among them, eight key candidate genes including serotonin receptor type 1B, crustacean hyperglycemic hormone, molt-inhibiting hormone, cytochrome P450

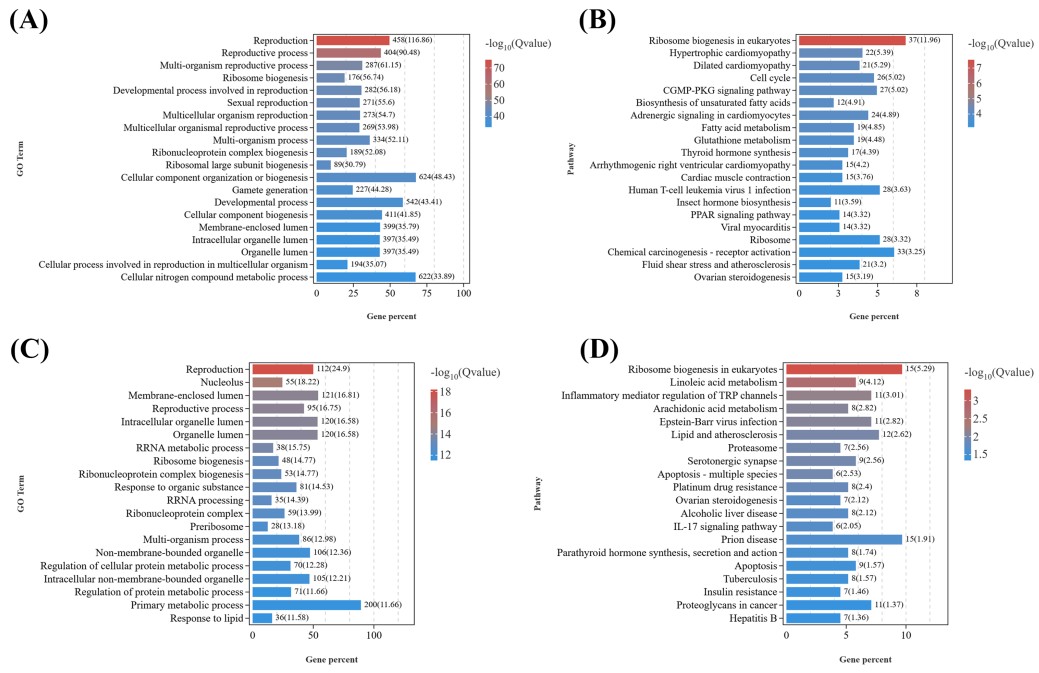

**Figure 5** **KEGG pathway enrichment analyses of DEGs related to reproduction.** (A) GO enrichment analysis of DEGs related to reproduction in females. (B) Top 20 KEGG pathways of DEGs related to reproduction in males. (C) GO enrichment analysis of DEGs related to reproduction in females. (D) Top 20 KEGG pathways of DEGs related to reproduction in females.

CYP2, cytochrome P450 2J4-like, homeotic protein distal-less-like, homeobox protein Hox-A6-like isoform X2, and abdominal-A homolog isoform X1 are down-regulated in parasitized males, while six key candidate genes including juvenile hormone esterase 1, juvenile hormone esterase binding protein, doublesex and mab-3 related transcription factor 3, truncated-like isoform X1, doublesex- and mab-3-related transcription factor 2-like, DNA methyltransferase 1, dachshund homolog 1-like are up-regulated. These DEGs are related to neuroendocrine regulation, sexual differentiation, and gonadal development in crustaceans.

Similarly, as female parasitized crabs exhibited gonadal atrophy and abnormal oocyte development, we performed GO analysis and KEGG analysis on reproduction-related DEGs (Data S9) from T2 *vs.* CK2 (Figs. 5C and 5D). The top 20 significantly enriched pathways included: ribosome biogenesis in eukaryotes (ko03008), linoleic acid metabolism (ko00591), serotonergic synapse (ko04726), proteasome (ko03050), ovarian steroidogenesis (ko04913), insulin resistance (ko04931), et al. DEGs were enriched in GO terms related to metabolic processes, signal transduction, reproductive processes, *etc*. Ten key candidate genes include doublesex- and mab-3-related transcription factor 2-like, cytochrome P450 CYP2, cytochrome P450 CYP2B, cytochrome P450 2J4-like, prostaglandin D synthase, prostaglandin E synthase, oxytocin/vasopressin-like peptide, pigment dispersing hormone, pigment-dispersing hormone 2 peptides-like, homeobox protein Hox-A6-like isoform X2.

**Table 3** DEGs related to reproductive metabolism.

| Gene ID | Name | Change trend in DEGs of CK1 VS T1 | Change trend in DEGs of CK2 VS T2 | Description |
|---|---|---|---|---|
| ncbi_126985833 | 5-HT1B | down | / | serotonin receptor type 1B |
| ncbi_127006217 | CHH | down | / | crustacean hyperglycemic hormone |
| ncbi_126987031 | MIH | down | / | molt-inhibiting hormone |
| ncbi_127004133 | JHE | up | / | juvenile hormone esterase 1 |
| ncbi_127003542 | JHEB | up | / | juvenile hormone esterase binding protein |
| ncbi_127000195 | Dmrt3 | up | / | doublesex and mab-3 related transcription factor 3, truncated-like isoform X1 |
| ncbi_126985409 | Dmrt2-like | up | down | doublesex- and mab-3-related transcription factor 2-like |
| ncbi_127003541 | Dnmt1 | up | / | DNA methyltransferase 1 |
| ncbi_126985078 | CYP2L1 | down | down | cytochrome P450 CYP2 |
| ncbi_127003389 | CYP2B | / | down | cytochrome P450 CYP2B |
| ncbi_126987539 | CYP2J4 | down | down | cytochrome P450 2J4-like |
| ncbi_127004463 | PGDS | up | down | prostaglandin D synthase |
| ncbi_126980964 | PGES | up | up | prostaglandin E synthase |
| ncbi_126996419 | OT/VP | down | up | oxytocin/vasopressin-like peptide |
| ncbi_126995690 | PDH1 | / | up | pigment dispersing hormone |
| ncbi_126993157 | PDH2 | / | up | pigment-dispersing hormone 2 peptides-like |
| ncbi_127008850 | Dac | up | / | dachshund homolog 1-like |
| ncbi_126998606 | Dll | down | / | homeotic protein distal-less-like |
| ncbi_127008919 | Antp | down | down | homeobox protein Hox-A6-like isoform X2 |
| ncbi_127008928 | Abd-A | down | / | homeobox protein abdominal-A homolog isoform X1 |

These critical reproduction-related DEGs were listed in Table 3. Some of these genes were enriched in the pathways discussed above, and some have been previously reported to potentially regulate male reproductive metabolism and gonadal development in crustaceans (*Levy & Sagi, 2020*; *Xu et al., 2022*; *Farhadi, 2023*; *Gao et al., 2023*).

## Validation of DEGs by qPCR

To validate the reliability of the transcriptomic data and further verify the expression changes of key genes involved in growth and reproduction regulation by *Polyascus* sp. parasitism, five target genes (*Dib*, *HR38*, *GST*, *CYP2L1*, and *PGDS*) along with the reference gene β-actin were detected with qPCR. As shown in Fig. 6, the regulation trends of these genes were consistent between qPCR and RNA-Seq results (Data S10), confirming the accuracy and reliability of the RNA-seq data.

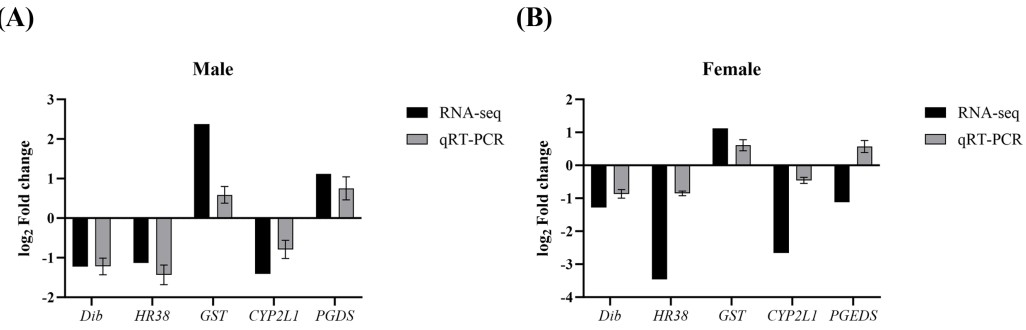

**Figure 6 Comparison of relative fold change of DEGs between RNA-seq and qRT-PCR.**
(A) Comparison in males. (B) Comparison in females.

## DISCUSSION

### Ontogeny and parasite-host interaction in *E. sinensis*

The salinity ranges for *E. sinensis* mating, spawning, egg brooding, embryonic development, and zoea larvae are 15–23, 6–14, 5–15, 10–16, and 6–20 (*Chunbo et al., 2023*), respectively, and the optimal salinity range for large-eyed larvae is 15–25. It has been reported that most of the maturation and larvae hatching of *Polyascus gregaria* in the Yangtze River estuary occur in waters with a salinity of 15–20 (*Geng, Zhang & Zhao, 2018*), which covers the waters where *E. sinensis* grows and reproduces. As rhizocephalans lack infectivity in freshwater systems, our collection of parasitized crabs from the Anqing freshwater section suggests initial infection likely occurred during estuarine residency as megalopa or early juveniles. Following infection, host crabs migrate upstream while the parasite develops its nutrient-absorbing root system (interna), significantly impairing host molting and growth dynamics. The host crabs caught during this research all had externas, indicating that externas were not completely dislodged because of their presence in the freshwater environment in the wild, proving that *Polyascus* sp. have a certain degree of adaptive ability in the freshwater environment. It is reported that reducing salinity of water can reduce the number of externas in hosts parasitized by *Polyascus gregaria* and control the spread of disease (*Xiang, Qianjun & Xiuyun, 2013*). It is supposed that *Polyascus* sp. may reduce the effect of the freshwater environment on them by changing their osmotic pressure regulating ability, inducing protection of their externas from shedding. However, this hypothesis needs to be verified by further study. The field investigation also revealed a substantial accumulation of sediment within the abdomen of the host crab, suggesting a decrease in the frequency of host activity or a diminished ability for self-cleaning. At the same time, the transcriptome analysis demonstrated that the common DEGs in both T1 *vs.* CK1 and T2 *vs.* CK2 comparison groups were significantly enriched in neurodegenerative disease-related pathways, including Alzheimer's disease (ko05010), Parkinson's disease (ko05012), Huntington's disease (ko05016), and neuroactive ligand–receptor interaction (ko04080). The decreased expression of *TUBA1A* (Tubulin Alpha Chain-like) and increased expression of various proteasome subunits (such as proteasome subunit alpha type-4-like and proteasome subunit alpha type-2-like) in these pathways suggested microtubule

damage, proteasome overactivation, protein homeostasis imbalance, and host motor neuron dysfunction. In addition to pathways mentioned above, common DEGs were also annotated to Dopaminergic synapse pathway (ko04728), which is reported to be closely related to neuroendocrine regulation (*Jungmann & Russell, 1977*; *Leslie & Nairn, 2019*), our result that reduced *CREB1* (cyclic AMP-dependent transcription factor ATF-1-like isoform X1) expression along with increased *PP2A* (protein phosphatase PP2A 55 kDa regulatory subunit isoform X2) and *PP1* (serine/threonine-protein phosphatase alpha-2 isoform-like isoform X1) expression in the dopaminergic synapse pathway indicated suppressed dopamine signaling, disrupted neuroendocrine regulation, and diminished behavioral activity in the host. This neural dysregulation and reduced host activity consequently decreased mechanical disturbance to the parasite's externa, thereby facilitating parasitic success (*Waiho et al., 2021*). Accordingly, we speculate that *Polyascus* sp. may also reduce the frequency of opening and closing of host abdomen by affecting the neuromuscular contraction ability of the host, thus reducing the contact between the ectoderms and the external water body. Based on our observations, we hypothesize that the *Polyascus* sp. employs dual adaptive strategies to enhance its survival in freshwater environments: On the one hand, the parasite may modulate its osmotic regulation capacity, and this physiological adjustment mitigates the detrimental effects of freshwater conditions. On the other hand, the parasite appears to interfere with host neuromuscular function, specifically reducing the frequency of abdominal flexions. Mechanistically, this may involve secretion of neuroactive compounds, modulation of host neurotransmitter systems and physical obstruction of muscle function (*Miroliubov et al., 2020*; *Waiho et al., 2021*; *Lianguzova et al., 2023*). These conjectures require further experimental verification. Previous studies have proposed that rhizocephalan parasites interact with the host's nervous system by secreting hormones and neurotransmitters through their rootlets to manipulate host behavior and physiological activities (*Miroliubov et al., 2020*; *Lianguzova et al., 2021*). Our findings provide molecular-level evidence that further elucidates these sophisticated interactions between rhizocephalan and their host's nervous system.

### *Polyascus* sp. inhibiting host growth and development by interfering with molting

The most prominent effects of rhizocephalan parasite infection on host crabs are manifested in external morphological alterations (*Alvarez, Hines & Reaka-Kudla, 1995*; *Walker, 2001*; *Li et al., 2011*; *Waiho et al., 2017*). In addition to the conspicuous feminization of male hosts, infected crabs typically exhibit significantly smaller body sizes compared to their healthy counterparts. Our preliminary survey data demonstrated that both male and female parasitized crabs had substantially lower average body weight and carapace width than uninfected individuals (*Jing et al., 2024*). Although serological analyses indicated reduced ecdysteroid levels in male hosts with no significant changes in females, these results only reflect the immediate hormonal status and potential developmental trends at the time of capture. It is known that growth and body size in crustaceans are intrinsically linked to molting cycles (*Hyde, Elizur & Ventura, 2019*; *Chen et al., 2019*). Previous studies have confirmed that the presence of rhizocephalan externa can disrupt the host's molting

process (*Waiho et al., 2021*). Therefore, it indicated that the infected crabs carrying externa collected from the freshwater Anqing section of the Yangtze River were likely already in a prolonged state of molting inhibition.

In summary, our results suggest that the primary mechanism by which *Polyascus* sp. inhibits host growth is through the suppression of the host's molting process (Fig. 7). The molting process in crustaceans is regulated by neuroendocrine mechanisms, primarily involving the modulation of ecdysteroid titers, secretion of new cuticle, degradation of old cuticle, regulation of molting behavior, and cuticle tanning (*Song et al., 2017*; *Zou, 2020*). Our results demonstrate that in male hosts parasitized by *Polyascus* sp., the expression of eyestalk neuropeptides MIH and CHH, which regulate ecdysteroid synthesis (*Chen et al., 2019*), was significantly reduced. In both male and female hosts, we observed upregulated expression of *Nvd* gene involved in cholesterol conversion and downregulated expression of *Dib* (*Shyamal et al., 2018*; *Benrabaa et al., 2023*; *Benrabaa & Mykles, 2025*), indicating disrupted regulation of ecdysteroid synthesis and consequent impairment of final molting hormone production following parasitic infection. Concurrently, we detected down-regulation of genes participating in critical molting regulatory processes, including chitin synthase (*chs-2*) for new cuticle secretion, chitinases (*cht3*, *cht5*, *cht10*) for old cuticle degradation, and nuclear hormone receptors (*HR38*, *HR3*, *HR4*) and *ftz-f1* for molting behavior regulation (*Song et al., 2017*; *Zhao et al., 2018*; *Zhang et al., 2021*; *Li et al., 2023*). Up-regulation genes in parasitized males, including *Ccap*, *NO*, and *Burs α*, reveal an effect on molting regulation. These molecular alterations collectively suggest substantial inhibition of essential molting processes such as cuticle formation and ecdysis in host crabs (*Song et al., 2017*), ultimately leading to impaired growth and development. The chitinase and chitin synthase system plays a pivotal role in coordinating the synthesis and degradation of arthropod exoskeletons, participating in crucial physiological processes including morphogenesis, nutrient digestion, and pathogen defense (*Kramer & Muthukrishnan, 1997*; *Liu et al., 2022*). *Takahashi & Matsuura (1994)* demonstrated that rhizocephalan parasitism suppresses molting in shore crabs: while infected females exhibited reduced molting frequency without changes in molt increment size, males showed decreases in both molting frequency and increment size. Our findings further revealed more pronounced differential expression of molting-related genes in male hosts compared to females parasitized by *Polyascus* sp., corroborating the observed greater morphological impact of parasitic infection on male crabs.

### *Polyascus* sp. interfering with male host reproductive development *via* "neuroendocrine disruption and reproductive gene regulation"

This study revealed that male hosts parasitized by rhizocephalan barnacles exhibited not only significantly smaller body size and mass compared to healthy individuals, but also showed marked atrophy or even complete disappearance of testes and accessory glands, accompanied by significantly reduced serum testosterone levels, indicating severe impairment of reproductive development by parasitic infection. According to the analysis of the research, the regulatory strategies employed by the parasite to manipulate male

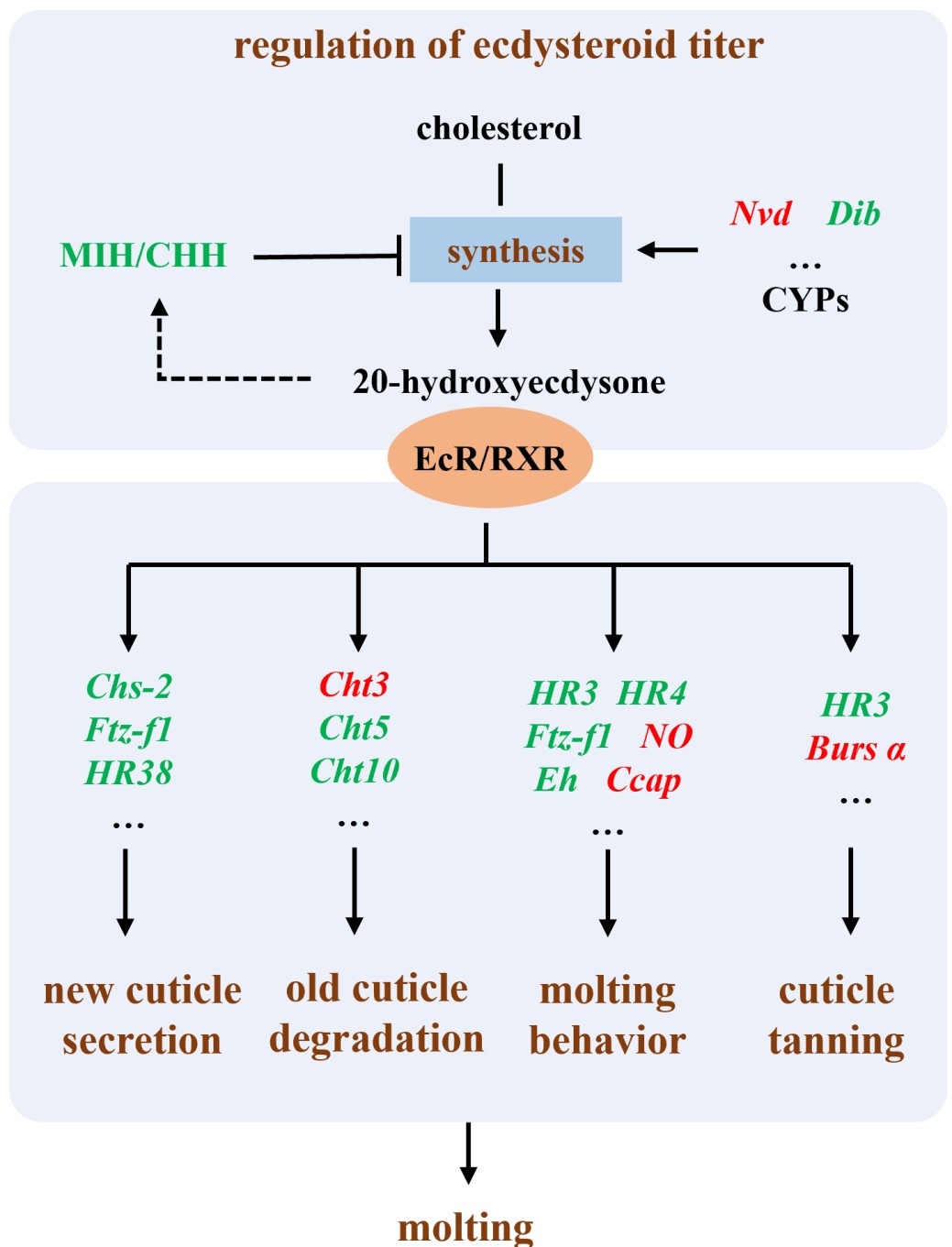

**Figure 7** Illustration of the molecular mechanism of molting interference by *Polyascus* sp.

host development can be summarized as two aspects: neuroendocrine disruption and reproductive gene regulation (Fig. 8).

Regarding neuroendocrine interference, existing studies have demonstrated that certain parasites (*e.g.*, nematodes and rhizocephalans) can secrete collagenase serine proteases to

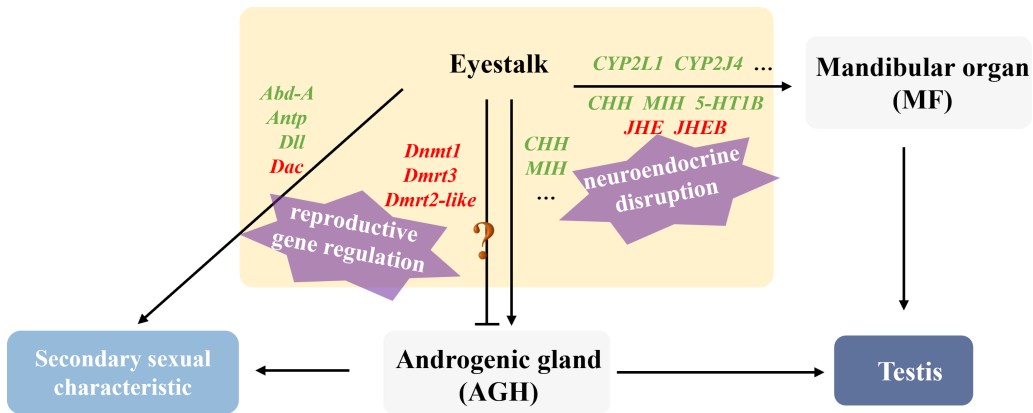

**Figure 8** Illustration of the molecular mechanism of "neuroendocrine disruption and reproductive gene regulation" in males parasitized by *Polyascus* sp. "?": it is unclear whether these genes are involved in the regulation of Androgenic gland development.

degrade collagen barriers (*e.g.*, basement membranes) in host tissues, facilitating invasion and nutrient absorption (*McKerrow et al., 1985*; *O'Brien et al., 2008*). We speculate that the parasite may interfere with serotonin receptor type 1B (*5-HT1B*) expression through two potential mechanisms: direct secretion of metalloproteinases to cleave receptor extracellular domains, or activation of host DNA methyltransferases (*Dnmt*) to induce methylation or chromatin condensation in the *5-HT1B* gene promoter region, ultimately suppressing its transcription (*Saki et al., 2020*). Downregulation of *5-HT1B* attenuates serotonin signaling, subsequently reducing stimulation to the mandibular organ (MO) and decreasing methyl farnesoate (MF) synthesis. MF is an insect juvenile hormone homologue (*Laufer et al., 2002*; *Olmstead & Leblanc, 2002*). It is reported that tissue-lytic enzymes secreted by the parasite may damage the host's MO, while combined with insufficient cholesterol uptake, collectively lead to reduced MF synthesis, ultimately causing gonadal developmental inhibition, testicular atrophy, and cheliped degeneration. Notably, proteases secreted by the parasite also damage eyestalk neurons, directly reducing CHH synthesis, while MF decrease further indirectly suppresses CHH secretion. *Martin et al. (2022)* identified several candidate genes in rhizocephalan interna, including immune-related genes, juvenile hormone esterase-binding protein (*JHEB*), and crustacean neurotoxins (*Li et al., 2025*). *JHEB* is likely derived from the parasite, and its upregulation antagonizes the function of *JHE*. In summary, the parasite significantly inhibits host serotonin signaling and eyestalk hormone (CHH, MIH) secretion, leading to reduced MF synthesis and energy metabolism disorders. Concurrently, the juvenile hormone metabolic system imbalance manifests as *JHE* accumulation, accelerating MF degradation, ultimately resulting in persistently low MF levels that directly suppress testis development. Cytochrome P450 is thought to be involved in the synthesis and metabolism of various steroid hormones in crustaceans (*James & Boyle, 1998*). The downregulation of *CYP2L1* and *CYP2J4* in the eyestalks of parasitized males suggests that the parasite may directly or indirectly affect male gonadal development by interfering with steroid hormone metabolism.

At the level of reproductive development regulation, *Polyascus* sp. systematically disrupts the Hox gene network and appendage development pathways of the host. As a crucial component of the retinal determination gene network (RDGN), the Dachshund homolog (*Dac*) participates in developmental regulation, cell differentiation, and organ formation across various species. In crustaceans, *Dac* likely modulates critical developmental events including appendage formation and eyestalk development (*Janssen et al., 2010*). The homeobox protein abdominal-A homolog isoform X1 (*Abd-A*) serves as a master regulator of abdominal development in crustaceans, ensuring proper morphogenesis and reproductive function by suppressing limb generation, specifying reproductive structures, and maintaining segment identity (*Averof & Akam, 1995*; *Hsia et al., 2010*). In healthy male individuals, *Abd-A* facilitates the specialization of abdominal locomotor appendages into reproductive limbs through developmental inhibition. The homeobox protein Hox-A6-like isoform X2 (*Antp*) regulates mid-hind thoracic segment development, primarily governing cheliped and walking leg formation in crabs, while also activating downstream target gene homeotic protein distal-less-like (*Dll*) to influence distal appendage development(*Pechmann et al., 2011*). Notably, analogous to *Toxoplasma gondii*'s strategy of secreting effector proteins to activate host Dnmt3a and methylate immune gene promoters, it indicated that *Polyascus* sp. may secrete specific proteins to upregulate host *Dnmt* expression, thereby methylating key male reproductive genes such as *Dmrt* to implement their ''demasculinization'' strategy. The results of our study demonstrate that *Polyascus* sp. parasitization significantly downregulates host *Abd-A* expression, which led to the degradation of the copulatory apparatus and even the growth of female individual-like ventral limbs with attached bristles. These morphological modifications may facilitate egg-brooding behavior, essentially protecting the parasite's externa. Concurrently, decreased expression of *Antp* in parasitized males causes subsequent downregulation of the downstream target gene *Dll*, ultimately manifesting as cheliped maldevelopment and atrophy. Collectively, the dysregulation of key developmental genes, including *Dmrt3*, Dmrt2-like, *Dnmt*, *Abd-A*, *Antp*, and *Dll* leads to characteristic alterations in male secondary sexual characteristics, such as reproductive appendage regression and cheliped atrophy.

### *Polyascus* sp. affecting female host reproductive development *via* "Nutrient Hijacking-Hormonal suppression" mechanism

In this study, the DEGs related to gonadal development in the T2 *vs.* CK2 comparison group were significantly enriched in pathways including ribosome biogenesis, linoleic acid metabolism, and ovarian steroidogenesis, indicating that *Polyascus* sp. parasitism substantially affects protein synthesis and fatty acid metabolism in female hosts. Notably, the male-specific DEGs (*5-HT1B*, *CHH*, *MIH*, *JHE1*, *JHEB1,* and *Dmrt3*) showed no significant changes in female crabs, while female hosts exhibited decreased expression of CYP450 family proteins and prostaglandin D synthase (*PGDS*). These findings suggest that *Polyascus* sp. does not comprehensively disrupt neuroendocrine systems in female hosts, but rather primarily exploit nutritional resources while maintaining the host's basic metabolic functions to ensure sustained energy supply (Fig. 9). Female crabs typically allocate substantial energy to vitellogenesis, whereas the parasite competes for critical resources

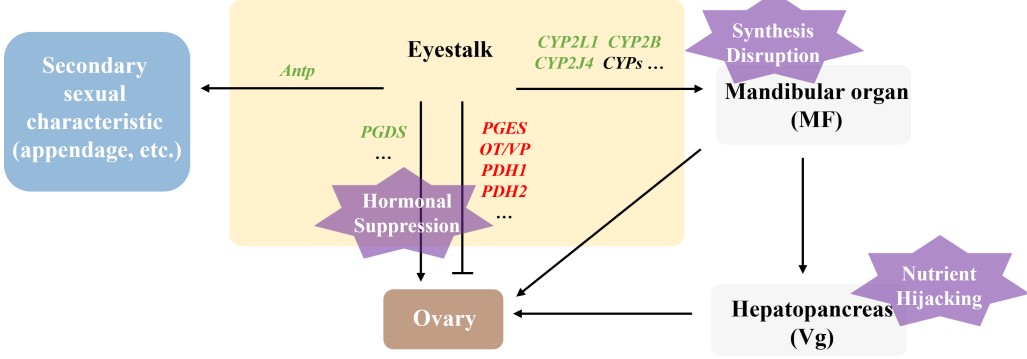

**Figure 9** Illustration of the molecular mechanism of "Nutrient Hijacking-Hormonal Suppression" in females parasitized by *Polyascus* sp.

like cholesterol and lipoproteins to support their development and reproduction. Upon infecting female hosts, *Polyascus* sp. damages the mandibular organ and depletes precursor cholesterol for MF synthesis. As MF normally activates vitellogenin (Vg) transcription in the hepatopancreas *via* nuclear receptor binding (*Chen et al., 2021*), its reduction inhibits Vg production. Concurrent hepatopancreas impairment further obstructs Vg synthesis, preventing yolk protein (Vn) accumulation in ovaries from reaching the threshold required for normal development, ultimately leading to ovarian atrophy. It is reported that neuropeptide pigment-dispersing hormone (PDH) decreases significantly during the mature stage, suggesting PDH is involved in regulating ovarian maturation in crustaceans (*Huang et al., 2014*). Our findings showed that *PDH1* and *PDH2* are up-regulated in parasitized females, which indicates the suppression of ovary development. Additionally, the parasite's interna absorbs host hemolymph cholesterol, causing CYP450 enzyme activity to decline due to substrate limitation. This disrupts steroid hormone (*e.g.*, sex hormones, ecdysteroids) synthesis and contributes to gonadal regression.

While female hosts showed reduced *Antp* expression level, *Abd-A*, *Dac*, and *Dll* expression remained unchanged, corresponding to morphological observations where cheliped development was impaired but abdominal morphology and secondary sexual characteristics exhibited minimal changes. The results indicate that although *Polyascus* sp. parasitism induces neuroendocrine and metabolic disturbances in female crabs, the parasite's reproductive strategy depends on the host's broad abdominal flap to protect its externa. Consequently, the evolutionary pressure to preserve female-specific morphological features may explain the limited impact on sexual differentiation compared to male hosts.

## CONCLUSION

In this study, morphological observations demonstrated that *E. sinensis* parasitized by *Polyascus* sp. in the freshwater habitats of the Yangtze River exhibited significant sex-specific alterations: males displayed pronounced feminization characteristics, while females showed relatively subtle phenotypic changes such as aberrant cheliped development. Notably, parasitized individuals of both sexes exhibited marked gonadal atrophy and

significantly smaller body sizes compared to the healthy individuals, indicating that *Polyascus* sp. parasitism profoundly impacts both morphological development and reproductive functions in *E. sinensis*. Comparative transcriptomic analysis of eyestalk tissues between parasitized and healthy crabs identified thirteen key candidate genes related to parasite-induced suppression of host molting and growth, fourteen crucial candidate genes involved in male reproductive regulation, and ten candidate genes affecting female gonadal development. These findings not only systematically elucidate the sexually dimorphic neuroendocrine manipulation strategies employed by the parasite, but also reveal the remarkable complexity of host-parasite interactions at the molecular level. The candidate gene clusters identified in this study provide important theoretical foundations and potential molecular targets for in-depth investigations into the mechanisms governing crustacean reproductive regulation and the development of anti-parasitic strategies.

### Funding

This study was supported by "Monitoring of aquatic resources in key waters of Anhui province" (2023AHNYC016XQ), and the Agricultural Finance Special Project "Central Public-interest Scientific Institution Basal Research Fund, CAFS" (NO. 2023TD11). The funders had no role in study design, data collection and analysis, decision to publish, or preparation of the manuscript.

### Grant Disclosures

The following grant information was disclosed by the authors:
Monitoring of aquatic resources in key waters of Anhui province: 2023AHNYC016XQ.
Agricultural Finance Special Project "Central Public-interest Scientific Institution Basal Research Fund, CAFS": NO. 2023TD11.

### Competing Interests

The authors declare there are no competing interests.

### Author Contributions

- Jing Xie conceived and designed the experiments, performed the experiments, analyzed the data, prepared figures and/or tables, authored or reviewed drafts of the article, and approved the final draft.
- Congping Ying analyzed the data, authored or reviewed drafts of the article, and approved the final draft.
- Zhen Tang performed the experiments, prepared figures and/or tables, and approved the final draft.
- Yanping Yang conceived and designed the experiments, authored or reviewed drafts of the article, and approved the final draft.
- Kai Liu conceived and designed the experiments, analyzed the data, authored or reviewed drafts of the article, and approved the final draft.

## Field Study Permissions

The following information was supplied relating to field study approvals (i.e., approving body and any reference numbers):

Field experiments were approved by Ministry of Agriculture and Rural Affairs, China (project number: 2023AHNYC016XQ)

## Data Availability

The raw data is available at Genome Sequence Archive: Bio Project PRJCA040566 (CRA026020).

## Supplemental Information

Supplemental information for this article can be found online at http://dx.doi.org/10.7717/peerj.20089#supplemental-information.

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
