# Peer review of "Eyestalk transcriptome analysis reveals sexually dimorphic host-parasite interactions: divergent molecular strategies of *Polyascus* sp. for reproductive and growth regulation in *Eriocheir sinensis"

_PeerJ, doi:10.7717/peerj.20089_

## Round 0.1 · original submission · Major Revisions

Dear Dr. Xie, I ask you to respond to the reviewers' comments. Many of the comments will require significant revisions to this manuscript. This will enhance the scientific value of your article. I would like to see a new version of this article approved for publication.

**Language Note:** PeerJ staff have identified that the English language needs to be improved. When you prepare your next revision, please either (i) have a colleague who is proficient in English and familiar with the subject matter review your manuscript, or (ii) contact a professional editing service to review your manuscript. PeerJ can provide language editing services - you can contact us at [email protected] for pricing (be sure to provide your manuscript number and title). – PeerJ Staff

Reviewer 1 ·

Basic reporting

The manuscript is well written in professional and unambiguous English. The figures are appropriately designed and informative.

Relevant literature is adequately cited, and the background provides sufficient context, particularly on rhizocephalan parasitism and host physiology.

Experimental design

The study design, comparing four groups of parasitized and non-parasitized males and females, is suitable for evaluating sex-specific effects.

The focus on eyestalk transcriptome is a strength, considering its role in neuroendocrine regulation in crustaceans.

However, the manuscript lacks clear information on which part of the eyestalk (e.g., retina, sinus gland, X-organ) was used for RNA extraction. This is critical for interpreting the expression of CHH and other neuropeptide genes, and should be clarified.

Validity of the findings

The DEG detection threshold (|log2FC| ≥ 2, FDR < 0.05) is statistically robust.

qPCR validation confirms the reliability of RNA-seq results.

While the proposed mechanisms involving neuroendocrine disruption (e.g., MF, CHH, MIH suppression) are plausible, direct evidence of causality is limited. Future functional studies (e.g., hormonal assays, gene knockdown) will be needed to support these conclusions.

Additional comments

This study provides novel insights into the sex-specific manipulation of growth and reproduction in Eriocheir sinensis by Polyascus sp., and significantly advances our understanding of host-parasite molecular interactions in decapods.

The identification of candidate genes and proposed regulatory pathways offers a solid foundation for future research on crustacean endocrinology and parasitology.

With minor revisions and clarifications, the manuscript would be suitable for publication in PeerJ.

Reviewer 2 ·

Basic reporting

The manuscript would benefit from content reorganization as the authors have a lot of data to present. The introduction needs to be reworded to orient the reader to the current status of research in this field. What previous RNAseq studies have been done? What tissues were used? What were the salient findings?

Subsections need to be more or less consistent between the Methods, Results, and Discussion sections. For example, two strategies/mechnisms - “neuroendocrine disruption and reproductive gene regulation” and “Nutrient Hijacking-Hormonal Suppression” - are mentioned in the Abstract, and then in Discussion section, but reader is not made aware about what is currently known about these two mechanisms in the Introduction section, and no results are presented in the Results section.

In the Results section, mostly pathway enrichment outcomes are presented. A number of results are mentioned for the first time in the Discussion section.

In the Discussion section, several statements lack references. So, it is hard to differentiate between what is the outcome of the current study and what is already known.

Experimental design

All good. No Comment.

Validity of the findings

Please see my comments in section 1.

Additionally, a number of previous findings were cited in these two sections in the Discussion,
Polyascus sp. interfering with male host reproductive development via “neuroendocrine disruption and reproductive gene regulation”, and,
Polyascus sp. impair female host reproductive development via “Nutrient Hijacking-Hormonal Suppression” Mechanism.

It is a lot of information to process, and for the uninitiated, it may be overwhelming. For the reader to understand what is already known and whether the current study supports existing findings, I suggest that the different genes and how they interact and influence each other should be presented as a figure. What the current study adds to the existing knowledge can then be highlighted in the same figure.

Annotated reviews are not available for download in order to protect the identity of reviewers who chose to remain anonymous.

---

## Round 0.2 · accepted · Accept

Dear Dr. Xie, I congratulate you on the acceptance of this article for publication.

Reviewer 1 ·

Basic reporting

The authors have addressed my comments. Therefore, I have no further comments.

Experimental design

The authors have addressed my comments. Therefore, I have no further comments.

Validity of the findings

The authors have addressed my comments. Therefore, I have no further comments.

Additional comments

The authors have addressed my comments. Therefore, I have no further comments.